# Predictors of non-adherence to antihypertensive medications: A cross-sectional study from a regional hospital in Afghanistan

**Muhammad Haroon Stanikzai**[1,2,3], **Mohammad Hashim Wafa**[4], **Essa Tawfiq**[5], **Massoma Jafari**[6], **Cua Ngoc Le**[1,2], **Abdul Wahed Wasiq**[7], **Bilal Ahmad Rahimi**[8], **Ahmad Haroon Baray**[3], **Temesgen Anjulo Ageru**[1,2], **Charuai Suwanbamrung**[1,2]*

1 Public Health Research Program, School of Public Health, Walailak University, Thai Buri, Thailand,
2 Excellent Center for Dengue and Community Public Health (EC for DACH), Walailak University, Thai Buri, Thailand, 3 Faculty of Medicine, Department of Public Health, Kandahar University, Kandahar, Afghanistan, 4 Faculty of Medicine, Neuropsychiatric and Behavioral science Department, Kandahar University, Kandahar, Afghanistan, 5 The Kirby Institute, UNSW Sydney, Sydney, Australia, 6 McMaster University, Hamilton, Ontario, Canada, 7 Faculty of Medicine, Department of Internal Medicine, Kandahar University, Kandahar, Afghanistan, 8 Department of Pediatrics, Faculty of Medicine, Kandahar University, Kandahar, Afghanistan

* Yincharuai@gmail.com

**Data Availability Statement:** The authors confirm that all data underlying the findings are fully available without restriction. All relevant data are

## Abstract

### Background

Non-adherence to antihypertensive medications (AHMs) is a widespread problem. Cardiovascular morbidity and mortality reduction is possible via better adherence rates among hypertensive patients.

### Objectives

This study aimed to assess the prevalence of non-adherence to AHMs and its predictors among hypertensive patients who attended Mirwais Regional Hospital in Kandahar, Afghanistan.

### Methods

A cross-sectional study using random sampling method was conducted among hypertensive patients, aged ≥18 years in Mirwais Regional Hospital at a 6-month follow-up between October and December 2022. To assess non-adherence to AHMs, we employed the Hill-Bone Medication Adherence scale. A value below or equal to 80% of the total score was used to signify non-adherence. A multivariable binary logistic regression model was used to identify predictors of non-adherence to AHMs.

### Results

We used data from 669 patients and found that 47.9% (95%CI: 44.1–51.8%) of them were non-adherent to AHMs. The majority (71.2%) of patients had poorly controlled blood

within the paper and its Supporting Information files.

**Funding:** This study was financially supported by the Excellent Center for Dengue and Community Public Health, School of Public Health [WU-COE-66-16], and the Walailak University. The funders had no role in study design, data collection and analysis, decision to publish, or preparation of the manuscript.

**Competing interests:** The authors have no conflict of interest.

pressure (BP). The likelihood of non-adherence to AHMs was significantly higher among patients from low monthly-income households [Adjusted odds ratio (AOR) 1.70 (95%CI: 1.13–2.55)], those with daily intake of multiple AHMs [AOR 2.02 (1.29–3.16)], presence of comorbid medical conditions [AOR 1.68 (1.05–2.67), lack of awareness of hypertension-related complications [AOR 2.40 (1.59–3.63)], and presence of depressive symptoms [AOR 1.65 (1.14–2.38)].

## Conclusion

Non-adherence to AHMs was high. Non-adherence to AHMs is a potential risk factor for uncontrolled hypertension and subsequent cardiovascular complications. Policymakers and clinicians should implement evidence-based interventions to address factors undermining AHMs adherence in Afghanistan.

## Introduction

Chronic illnesses have constantly presented substantial health issues on a worldwide scale. Among these conditions, hypertension stands out as a particularly concerning disease. World Health Organization (WHO) estimated that ~1.28 billion adults aged 30–79 years had hypertension in 2019, and it may rise to 1.56 billion by 2025 [1–3]. According to the report, the majority (two-thirds) of hypertensive patients live in low- and middle-income countries (LMICs) [3]. Hypertension is a crucial contributor to cardiovascular complications and health loss—for instance, only in 2019, it claimed ~7.5 million deaths (12.8% of global deaths) [1–3]. The occurrence of geopolitical instability and economic downturns in certain places, such as Afghanistan, has had a dual impact on healthcare services, as well as the exacerbation of stress-related diseases, notably hypertension [4]. In 2020, 5.3 million hypertensive patients and 7,995 hypertension-linked deaths were reported in Afghanistan [4].

Worldwide, only 21% of hypertensive patients have their blood pressure under control [3,5,6]. The rate of blood pressure (BP) control differs between countries. In resource-poor countries such as India, Ethiopia, and Uganda, the level of controlled BP among treated patients was 17.5% [7], 47.9% [8], and 18% [9], respectively. In Afghanistan (with its healthcare system still recovering from years of conflict), the national prevalence of uncontrolled hypertension is ~80%, warranting immediate attention [4,10].

The benefits of lifestyle changes and pharmacotherapy in controlling blood pressure levels are well established [1,5]. However, low or no adherence to AHMs is a widespread problem. Aside from the obvious health consequences, poor adherence to AHMs has socioeconomic implications in developing countries, such as higher medical costs, missed workdays, and demand for healthcare systems [6]. Globally, AHMs adherence has recently become a central concern to clinicians, policymakers, and researchers. In Afghanistan, where access to healthcare services and medications is quite limited, AHMs-adherence acquires additional importance [4].

Adherence to AHMs varies globally, influenced by sociodemographic factors and lifestyle choices [11–18], leading to marked regional disparities in adherence levels. Moreover, adherence to AHMs is suboptimal, particularly in the developing world [9,11–15]. For instance, a recent study involving 27 million populations found that the global prevalence of no-adherence to AHMs was high, ranging from 27% to 40% [11]. A study from 22 Asian countries

reported that approximately 48% of hypertensive patients had poor adherence to their AHMs [12]. The proportion of hypertensive patients with low compliance to AHMs was 37.7% in Pakistan [13], 49.6% in Iran [14], 39.5%% in Nigeria [15], 34.5% in Ethiopia [16], 67.7% in Cameron [17], 37.4% in Nepal [18], and 4%-81% in India [19]. In 2023, Baray et al. showed that Afghanistan's current proportion of hypertensive patients with low adherence to hypertension treatment is 42.1% [10]. In Afghanistan, the high prevalence of uncontrolled hypertension compounded with non-adherence to AHMs constitutes a significant public health problem.

A fair amount of data in the developing world indicate that diverse factors such as age [11,20], sex [11,21,22], marital status [11,23], employment status [11,24], area of residence [11,20], education [11,25], health facility distance [21], household income [11,22], Body Mass Index (BMI) [23,24], physical activity [11,20], alcohol consumption [22], smoking [11], comorbidity [11,20–22], salt intake [11,20], family history of hypertension [11], hypertension duration [11], number of AHMs [11,20–22], stage of the disease [11], knowledge about hypertension and its treatment [21,24,25], history of depressive symptoms [11,23,24,26], hypertension-linked cost [11,22,24,25], social support [23,26], and client satisfaction [25] are associated with adherence to AHMs.

Although non-adherence to AHMs is a global health concern, in developing countries such as Afghanistan, where literacy, economy, and healthcare systems, to name a few, are highly devastated, it is even more worrisome. Non-adherence to AHMs is associated with poor blood pressure control, which, in turn, potentially contributes to the development of uncontrolled hypertension. A study conducted in Afghanistan indicates that 77.3% of hypertensive patients had poor BP control [10]. The literature gap, however, does exist on the prevalence of non-adherence to AHMs and its predictors in Afghanistan. Considering the gap in knowledge, we aimed to assess the prevalence of non-adherence to AHMs and related predictors among hypertensive patients who attended Mirwais Regional Hospital in Afghanistan to aid and direct future interventions.

## Materials and methods

### Study setting and design

We conducted this cross-sectional study at Mirwais Regional Hospital (MRH) in Kandahar City between October and December 2022. MRH is the largest public hospital allocated to healthcare provision in the southwest region of Afghanistan. MRH provides services primarily to the residents of 6 provinces (Kandahar, Helmand, Zabul, Oruzgan, Nimroze, and Farah), that is, more than 5000,000 population in its catchment area. MRH is a public hospital and provides secondary and tertiary health services free of charge to patients, including hypertensive patients. Thus, the data generated from this hospital may be somewhat considered a fair representation of the whole hypertensive mass of the country.

### Study population

Our target population consisted of any hypertensive patient, aged $\geq$ 18 years, who consented to participate in this study and was receiving antihypertensive follow-up treatment at MRH for at least six months at the time of data collection. We excluded pregnant women, unconscious and critically ill individuals, and currently hospitalized patients.

### Sample size and sampling procedure

The sample size for this study was estimated using the single population proportion formula, which accounts for the expected prevalence, desired confidence level, and allowable error. We

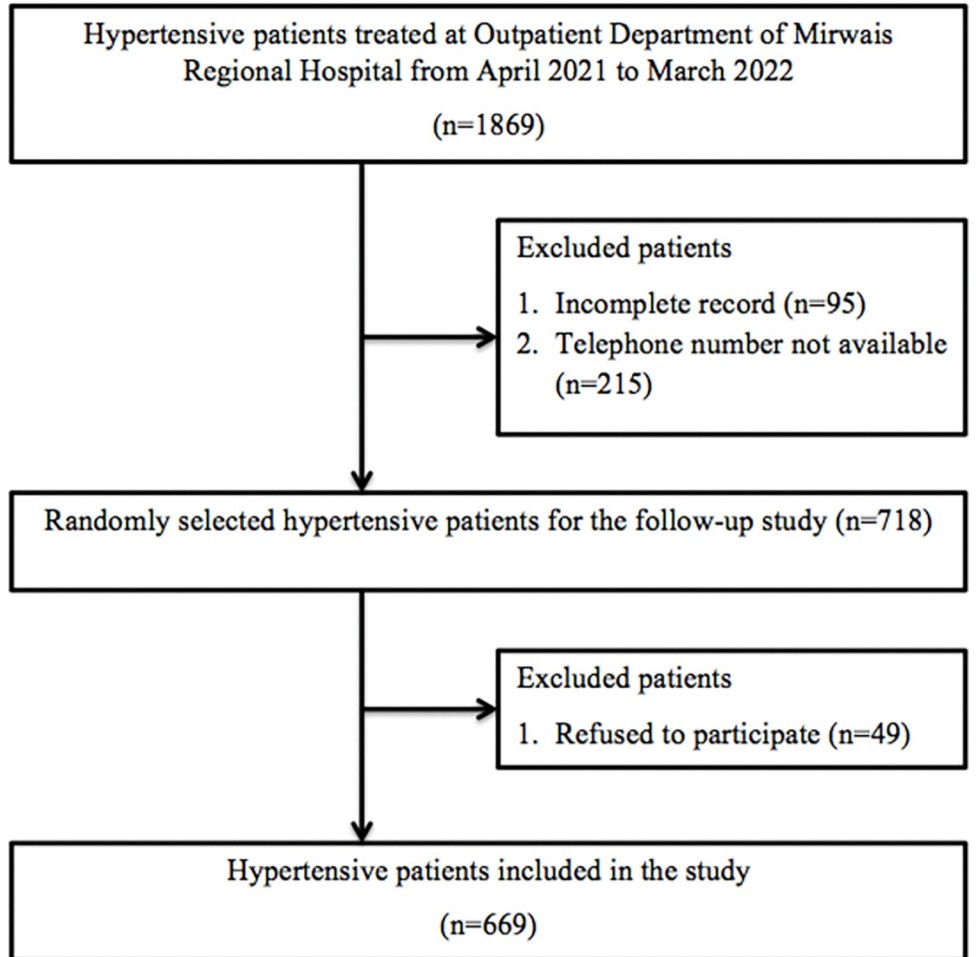

**Fig 1. Flowchart of participants' recruitment.**

employed a 42.1% prevalence rate of non-adherence to AHMs for calculating our sample size [10]. Considering 95% CI, 5% margin of error, 1.5 design effect, and a 15% non-response, we reached out to 718 hypertensive patients. Out of those contacted, 669 agreed to join the study while others declined or were unavailable. The sampling procedure involved a random selection of hypertensive patients who received antihypertensive treatment at the outpatient department (OPD) of the hospital six months prior to data collection. We accessed the total list of our hypertensive subjects from the patients' registration book of the respective hospital in September 2022, consisting of 1869 hypertensive patients. The data pertaining to patients with hypertension was recorded in Microsoft Excel spreadsheets. We employed a random sampling procedure to select the 718 respondents for our follow-up study (Fig 1).

## Study variables

**Dependent variable.** The outcome variable was the proportion of patients with non-adherence levels to AHMs on the Hill-Bone Medication Adherence Scale. The Hill-Bone Medication Adherence Scale, designed by Johns Hopkins University (1999), contains nine items [27]. Each item is a four-point scale ranging from 1 (all the time) to 4 (none of the time), yielding a total score from 9 to 36. A value below or equal to 80% of the total score was used to

signify medication non-adherence [28–30]. The Hill-Bone Medication Adherence Scale is a globally credible instrument with good psychometric properties for assessing medication adherence across various chronic diseases and conditions [30,31]. The Cronbach's alpha value for the Pashtu version was 0.89. This scale has been used previously in Afghanistan and has demonstrated excellent psychometric properties [10].

**Independent variables.** Socio-demographic factors included age, marital status, sex, residence, education, employment, household monthly income, household size, and Body Mass Index (BMI).

Behavioral factors included smoking history, level of physical activity, vegetable and fruit consumption, and added table salt intake.

Disease-related factors included family history of high BP, duration of hypertension, presence of comorbid medical conditions, hospitalization history in the last six months, nonsteroidal anti-inflammatory drugs (NSAIDs) use, BP monitoring at home (measurement of BP at home), number of AHMs, and source of AHMs (public vs. private pharmacies).

In this study, we measured BP using a new manual sphygmomanometer, following the guidelines set by the American Heart Association [32]. The device was tested on several patients, and its readings of systolic BP and diastolic BP were compared with several other new sphygmomanometers. We found no major differences between the readings of BP between the sphygmomanometer we used and others. All patients had their BP measured twice, each separated 10 minutes apart. The average of two BP values was used in the analysis. BP was defined as controlled if systolic BP was <140 mmHg and diastolic BP was <90 mmHg or uncontrolled if either one or both were elevated [33].

We employed the 9-item self-administered Patient Health Questionnaire (PHQ-9) scale to assess depressive symptoms in our patients over the last two weeks [34]. All nine items of PHQ-9 were scored on a 4-point Likert scale from 0 (not at all) to 3 (nearly every day), yielding a total score from 0 to 27. The total scores were grouped into different categories of depression symptoms based on the following ranges: normal = 1–4, mild = 5–9, moderate = 10–14, severe = 15–19, and extremely severe = 20–27 [34,35]. The PHQ-9 demonstrated an excellent internal reliability with a Cronbach's α of 0.92. The PHQ-9 has been used previously in Afghanistan [10].

Additionally, we employed the International Physical Activity Questionnaire short form (IPAQ-SF) to assess physical activity levels in our population [36]. We considered walking 3 days/week for at least 20 min/day to be low level of physical activity or < 3 metabolic equivalents (METs). Moderate physical activity 3-5days/week for at least 30 min/day represented medium level of physical activity or 3 to 7.9 METs. Vigorous physical activity 3–7 days/week signified high level of physical activity or >8.0 METs [36,37]. The Cronbach's α value for our sample was 0.876.

## Data collection

The questionnaire, which consisted of sections on socio-demographic information, behavioral factors, disease-related information, IPAQ-SF, PHQ-9, and the Hill-Bone Medication Adherence Scale, was initially drafted in English and later translated into the local language (Pashtu) for the ease of administration. Before the commencement of the study, we pretested the questionnaire in another setting (Kandahar Teaching Hospital) with 59 participants to check and revise (if required) its verbal consistency.

Two male and two female doctors with an MD degree in curative medicine and a minimum of three years of clinical experience composed our interview team. Although some interviewers had participated in other clinical studies, all interviewers received one-day training for this

study. To ensure cultural sensitivity and patient comfort, particularly given the diverse demographic of hypertensive patients, we strategically included two male and two female doctors in the team. The investigators screened all records of hypertensive patients registered in the OPD of MRH from April 2021 to March 2022. We employed a random selection process and made a telephone call to 718 patients. If the patients would like to participate, they could willfully provide verbal consent and schedule the interview at their convenience. Female doctors interviewed female patients. This gender division facilitated smoother interactions, especially in scenarios where female patients preferred being interviewed by a female doctor. Each interview took approximately 20 minutes to complete. We called consented patients from October to December 2022, and the principal investigators supervised the data collection process. Participants were compensated for their travel expenses and time. We checked the questionnaires within 24 hours for completion.

## Statistical analysis

We transferred the data from Microsoft Excel 2019 to IBM SPSS Statistics version 21.00 for cleaning and analyses [38]. We employed descriptive statistics for most variables, such as frequency and percentage. Next, we conducted univariate and multivariable analyses to identify predictors of non-adherence. We adjusted the model for potential confounders such as age and sex. We set the significance level at a $P$ value of $<0.05$.

## Ethical approval

The permission to conduct the study was obtained from the Public Health Directorate of Kandahar Province and the Research and Ethics Committee (Faculty of Medicine, Kandahar University) approved this study (Certificate # 17, Dated 15/July/2022). We obtained informed consent either in written form or in oral form (if the participant was illiterate). We declare that we carried out all methods in light of relevant guidelines and regulations. We also assert that all procedures contributing to this work adhere to established guidelines for medical research involving human subjects and with the Helsinki Declaration of 1975, as revised in 2008.

## Results

A total of 669 patients with hypertension were enrolled in the present analyses (response rate, 93.1%). The mean age (±SD) was 47.5 (± 9.62) years. Of them, 52% (348) were male, and 53.5% (358) were urban residents. Moreover, 606 (90.6%) were married, 449 (67.1%) had no formal education, and 421 (62.9%) were unemployed. The monthly median average household income was 9000 Afghani (IQR: 6000–13,000), which is equivalent to approximately USD 100 (August 2023). Their mean BMI (±SD) was 23.3 (±3.36), and about a fourth (23.8%) of them were overweight/obese. Table 1 depicts the detailed socio-demographic characteristics of our participants.

The majority (79.5%; 532) of them were physically inactive, 28.9% (193) were current cigarette smokers, and 33.0% (221) reported adding salt to their food at the table. Table 2 summarizes the behavioral characteristics of our participants.

The median duration of hypertension diagnosis was 6.88 years (IQR 3.4–12.1 years), and nearly half of the patients (47.5%, 318) had a positive family history of hypertension. A large percentage (38.9%) of the patients had some type of comorbid health condition, and one-third (32.6%, 218) had no awareness of hypertension-related complications. Of all treated hypertensive patients, 24.8% (166) monitored their BP at home, 38.7% (259) were on ≥3 AHMs, and 71.2% (476) had poor control of their BP. Overall, 345 (51.6%) patients had depressive symptoms, including 70 (10.5%) mild, 183 (27.4%) moderate, 89 (13.3%) severely moderate, and 3 (0.4%) severe cases (Table 3).

**Table 1. Socio-demographic characteristics of the study participants (n = 669).**

| Variables | Frequency (%) |
|---|---|
| **Age (In completed years)** | |
| 18–29 | 16 (2.4) |
| 30–39 | 98 (14.6) |
| 40–49 | 297 (44.4) |
| 50–59 | 186 (27.8) |
| ≥ 60 | 72 (10.8) |
| **Sex** | |
| Male | 348 (52.0) |
| Female | 321 (48.0) |
| **Residence** | |
| Urban | 358 (53.5) |
| Rural | 311 (46.5) |
| **Marital status** | |
| Single | 15 (2.3) |
| Married | 606 (90.6) |
| Widowed | 3 (0.4) |
| Divorced | 45 (6.7) |
| **Educational Status** | |
| No formal education | 449 (67.1) |
| Primary | 73 (10.9) |
| Secondary | 80 (12.0) |
| High school graduate | 54 (8.1) |
| Higher studies | 13 (1.9) |
| **Employment stauts** | |
| Self-employed | 191 (28.6) |
| Public employed | 34 (5.1) |
| Private/NGO employed | 23 (3.4) |
| Housewife | 216 (32.3) |
| Unemployed | 205 (30.6) |
| **Household members** | |
| ≤8 | 359 (53.7) |
| >8 | 310 (46.3) |
| **Monthly household income (in Afghanis)** | |
| ≤ 10000 | 446 (66.7) |
| 11000–20000 | 164 (24.5) |
| > 21000 | 59 (8.8) |
| **BMI status** | |
| Under weight | 5 (0.7) |
| Normal weight | 505 (75.5) |
| Over weight | 113 (16.9) |
| Obese | 46 (6.9) |

**Abbreviations:** BMI, Body Mass Index; NGO, Non-Governmental Organizations.

Out of 669 participants, 341 were non-adherent to their AHMs according to the Hill-Bone Medication Adherence scale, giving a non-adherence prevalence of 47.9% (95%CI: 44.1–51.8%).

Table 4 lists the results of univariate and multivariable analyses. For each independent variable in the table, the category marked with "1" represents the reference category against which odds ratios for other categories are compared. After controlling for age and sex, the likelihood of non-adherence to AHMs was significantly higher for patients living in households with an income of <10000 Afghanis/month (AOR = 1.70: 1.13–2.55). For clinical characteristics, daily intake of multiple AHMs (AOR = 2.02: 1.29–3.16), presence of comorbid medical conditions

**Table 2. Behavioral characteristics of the study participants (n = 669).**

| Variables | Frequency (%) |
|---|---|
| **Current cigarette smoking** | |
| Yes | 193 (28.9) |
| No | 476 (71.1) |
| **Physical activity** | |
| Yes | 137 (20.5) |
| No | 532 (79.5) |
| **Level of physical activity (n = 137)** | |
| Low | 33 (24.1) |
| Moderate | 70 (51.1) |
| High | 34 (24.8) |
| **Fruits consumption** | |
| Rarely | 313 (46.8) |
| Sometimes | 330 (49.3) |
| More often | 26 (3.9) |
| **Vegetables consumption** | |
| Rarely | 273 (40.8) |
| Sometimes | 321 (48.0) |
| More often | 75 (11.2) |
| **Meat consumption** | |
| Rarely | 522 (78.0) |
| Sometimes | 85 (12.7) |
| More often | 62 (9.3) |
| **Added table salt** | |
| Yes | 221 (33.0) |
| No | 448 (67.0) |

**Notes:** Rarely (less than once a week); Sometimes (1–3 times a week); More often (4–7 times a week).

(AOR = 1.68: 1.05–2.67), lack of awareness of hypertension-related complications (AOR = 2.40: 1.59–3.63), and presence of depressive symptoms (AOR = 1.65: 1.14–2.38) were significantly associated with higher of odds of non-adherence to medications (Table 4).

## Discussion

This study assessed hypertension medications non-adherence and its predictors among hypertensive patients attending a regional hospital at the six months follow-up in southwest Afghanistan. We found that 47.9% of hypertension patients were non-adherent in their AHMs use. Factors that mainly led to non-adherence were low monthly household income, antihypertensive regimens requiring multiple medications, presence of comorbid medical conditions, lack of awareness of hypertension-related complications, and presence of depressive symptoms. Additionally, the majority (71.2%) of the patients had poor control of their BP.

We found that 47.9% of our subjects were non-adherent for AHMs, with a 95% CI of 44.1% to 51.8%. This figure is higher than the non-adherence rates claimed in many studies conducted in other developing countries, such as Pakistan, Nigeria, Ethiopia, and Nepal [13,15,16,18]. Therefore, our findings signify an outstanding example of an LMIC country with limited resources and lagging behind the adherence rates to AHMs of the developed world [14,18]. Since non-adherence to AHMs inevitably predicts uncontrolled hypertension and subsequent cardiovascular complications, authorities should plan interventions to improve medication adherence among hypertensive patients. Such interventions could include

**Table 3. Disease related characteristics of the study participants (n = 669).**

| Variables | Frequency (%) |
|---|---|
| **Duration of hypertension (years)** | |
| 1–5 | 311 (46.5) |
| $\geq$ 5 | 358 (53.5) |
| **Family history of hypertension** | |
| Yes | 318 (47.5) |
| No | 351 (52.5) |
| **Comorbid medical disease** | |
| Yes | 260 (38.9) |
| No | 409 (61.1) |
| **BP monitoring at home** | |
| Yes | 166 (24.8) |
| No | 503 (75.2) |
| **NSAIDs use** | |
| Yes | 205 (30.6) |
| No | 464 (69.4) |
| **Number of AHMs** | |
| Monotheraphy | 190 (28.4) |
| Dual therapy | 220 (32.9) |
| $\geq$ 3 | 259 (38.7) |
| **Hospitalization in the last six months** | |
| Yes | 135 (20.1) |
| No | 534 (79.9) |
| **Source of AHMs** | |
| Private pharmacies | 607 (90.7) |
| Public | 62 (9.3) |
| **BP level** | |
| Controlled | 193 (28.8) |
| Uncontrolled | 476 (71.2) |
| **Knowledge about hypertension complications** | |
| Yes | 451 (67.4) |
| No | 218 (32.6) |
| **Severity of depression (PHQ-9)** | |
| None/minimal depression | 324 (48.4) |
| Mild depression | 70 (10.5) |
| Moderate depression | 183 (27.4) |
| Moderately severe depression | 89 (13.3) |
| Severe depression | 3 (0.4) |

**Abbreviations:** BP, Blood Pressure; NSAIDs, Non-Steroidal Anti-Inflammatory Drugs; AHMs, Anti-Hypertensive Medications; PHQ-9, Patient Health Questionnaire-9.

patient education, simplifying medication regimens, implementing reminder systems, and encouraging regular follow-up appointments.

Consistent with relevant literature, we found that patients with low monthly household incomes were more susceptible to non-adherence to AHMs. Moreover, private pharmacies provided medicines for a vast majority (90.7%) of the patients. Similar studies have shown that low income could interfere with good medication adherence among hypertensive patients [11,22,24]. It may epitomize the difficulty low-income patients encounter while bearing the economic burden of especially long-term hypertension treatment. Furthermore, the harsh economic conditions in the country might plunge many hypertensive patients into poverty. Therefore, low-income patients require adequate financial support from family members by

**Table 4. Factors associated with non-adherence to AHMs; crude and adjusted odds ratio with 95% CI.**

| Independent Variables | Categories | Non-adherence | | COR (95% CI) | P-Value | AOR (95% CI) | P-Value |
|---|---|---|---|---|---|---|---|
| | | Yes | No | | | | |
| Age | < 40 | 49 | 65 | 1 | 0.21 | - | - |
| | ≥40 | 272 | 283 | 1.27 (0.84–1.91) | | | |
| Marital status | Currently married | 286 | 320 | 1 | 0.20 | - | - |
| | Currently unmarried | 35 | 28 | 1.35 (0.84–2.17) | | | |
| Employment | Employed | 210 | 243 | 1 | 0.21 | - | - |
| | Unemployed | 111 | 105 | 1.22 (0.88–1.69) | | | |
| Monthly household income | <10000 | 241 | 205 | 2.10 (1.50–2.92) | <0.001 | 1.70 (1.13–2.55) | 0.01 |
| | ≥10000 | 80 | 143 | 1 | | 1 | |
| Physical activity | Yes | 47 | 90 | 1 | <0.001 | - | - |
| | No | 274 | 258 | 2.03 (1.37–3.08) | | | |
| Smoking | Yes | 102 | 91 | 1.31 (0.94–1.83) | 0.10 | - | - |
| | No | 219 | 257 | 1 | | | |
| Added table salt | Yes | 120 | 101 | 1.46 (1.05–2.01) | 0.02 | - | - |
| | No | 201 | 247 | 1 | | | |
| Duration of hypertension | 1–5 years | 125 | 186 | 1 | <0.001 | - | - |
| | > 5 years | 196 | 162 | 1.80 (1.32–2.44) | | | |
| Number of medications | Single | 67 | 123 | 1 | <0.001 | 1 | 0.02 |
| | Multiple | 254 | 225 | 2.07 (1.46–2.93) | | 2.02 (1.29–3.16) | |
| Comorbid disease | Yes | 134 | 126 | 1.51 (1.03–2.22) | 0.03 | 1.68 (1.05–2.67) | 0.01 |
| | No | 187 | 222 | 1 | | 1 | |
| Hospitalization in the last six months | Yes | 76 | 59 | 1.26 (0.92–1.72) | 0.06 | - | - |
| | No | 245 | 289 | 1 | | | |
| Family history of hypertension | Yes | 164 | 154 | 1.31 (0.97–1.78) | 0.07 | - | - |
| | No | 157 | 194 | 1 | | | |
| Knowledge of hypertension complications | Yes | 201 | 250 | 1 | 0.01 | 1 | <0.001 |
| | No | 120 | 98 | 1.52 (1.1–2.10) | | 2.40 (1.59–3.63) | |
| Depression | Yes | 196 | 149 | 2.09 (1.53–2.85) | <0.001 | 1.65 (1.14–2.38) | 0.007 |
| | No | 125 | 199 | 1 | | 1 | |

**Abbreviations:** COR, Crude Odds Ratio; AOR, Adjusted Odds Ratio, CI, Confidence Interval; BP, Blood Pressure; AHMs, Anti-Hypertensive Medications.

providing money for drugs and transportation. In addition, government or non-profit organizations can offer assistance programs to help low-income patients access affordable medications.

Our finding of the positive association between non-adherence and daily intake of multiple AHMs is consistent with studies from Korea [20], Malaysia [39], Lebanon [23], and elsewhere [11,18,21,24]. The everyday use of multiple daily medications is an essential intervention to decrease cardiovascular complications in patients with poorly controlled hypertension. Also, their benefits in treating hypertensive patients with comorbidities have been shown in the literature [40,41]. However, regimens with more than one medication per day are inconvenient, and as a consequence, patients find them difficult to follow, and potentially compromising the effectiveness of the treatment. Hence, the combination and preparation of the two AHMs as one tablet, known as a fixed-dose combination (FDC), might improve patient adherence [21,41]. Before prescribing FDCs, healthcare providers should carefully consider the medical history of patients, their current health status, and potential drug interactions. It is also worth mentioning that the presence of side effects from multiple medications and limited financial resources may also contribute to non-adherence in such cases.

The association between comorbidity and risk of non-adherence to AHMs is well documented. We found that patients with a comorbid medical condition were 1.6 times more likely

to be non-adherent to AHMs than their counterparts with no comorbidity. The increased difficulty of treating numerous illnesses simultaneously sometimes may cause patients to prioritize one over another or to feel overwhelmed, affecting their adherence. Similar findings from Korea [20] and Ethiopia [21] revealed comorbidity as a valid factor in tackling adherence. A possible reason for this could be a complicated treatment regimen. Besides, concomitant prescription of several medications prescribed for both hypertension and comorbidities that might result in a pill burden that inevitably leads to non-adherence. Another limitation associated with comorbidities is the potential for drug-drug interactions, which can hinder medication adherence. It is crucial for healthcare providers to effectively communicate the risks and benefits of each medication, especially when potential interactions exist [21]. By understanding the significance, patients may be more motivated to adhere to the prescribed regimen and report any unusual side effects promptly [21,40]. Therefore, hypertensive patients with a comorbidity warrant adequate care, supervision and counseling during their treatment.

Lack of awareness about hypertension complications showed an association with non-adherence to AHMs. This finding aligns with similar studies conducted in Eastern Ethiopia [25], Nigeria [42], and Congo [43]. Crucially, while this study did not provide gender-disaggregated data concerning the role of education in adherence, we must highlight the profound implications of the recent ban on education for Afghan women. Such bans can severely limit women's access to critical health information and potentially compound their vulnerability to diseases due to a lack of awareness. Given the socio-cultural landscape of Afghanistan, it is likely that women are disproportionately affected by such bans, leading to heightened risks regarding medication non-adherence. A lack of good knowledge about the nature of the disease might affect the patients' motivation. Educational interventions that provide patients with clear information about their condition, its progression, the potential risks of non-adherence, and the benefits of following the treatment plan can significantly improve medication adherence and overall disease management.

Depression is common in patients with hypertension and is associated with a poorer prognosis and higher healthcare costs [44,45]. As previously reported in other studies [11,23,24,26], the presence of depressive symptoms was associated with non-adherence to AHMs. Given the high prevalence of depression in Afghan society, this effect is highly relevant [46]. The relationship between mental and physical health is complex. Addressing mental health concerns such as depression is important for the patient's general well-being and may also be a catalyst for increasing adherence to physical health medical treatments [23,24]. Hence, screening and early identification of hypertensive patients with depressive symptoms are paramount to circumvent non-adherence to AHMs and their bio-psycho-social sequelae.

In this study, we found that 71.2% of the patients had poor control of their BP. This finding is consistent with previous studies conducted in Afghanistan [4,10] and other developing countries [7–9]. There are several examples in the literature that non-adherence to AHMs also plays a significant role in the suboptimal control of hypertension [9–11]. In Afghanistan, the high prevalence of uncontrolled hypertension compounded with poor adherence to AHMs raises causes for concern and action.

## Study limitations

We acknowledge the limitations of this study as follows: First, the cross-sectional nature of our study means that causal relationships between the reasons for non-adherence and the outcomes can't be firmly established. Second, our reliance on self-reported medication non-adherence might introduce errors due to recall biases or participants' desire to be viewed favorably, also known as social desirability bias. Third, we have not assessed the complex

interplay of social and cultural factors that can influence medication adherence in Afghanistan. Fourth, the unknown prevalence of non-adherence to AHMs among hypertensive patients lost to follow-up may have introduced bias in the analysis of predictors for non-adherence. Fifth, we lack knowledge regarding the type of AHMs our participants were taking for their medical conditions, which could affect the degree of non-adherence and the particular causes of non-adherence. Sixthly, our data epitomize a single regional hospital; therefore, making any generalization calls for caution. Finally, our results reflect a deficiency of assessing diverse levels (low, medium, or high) of medication non-adherence.

## Conclusion

About half of hypertensive patients, in our cohort, failed in their AHM adherence. Non-adherence to AHMs is a potential risk factor for uncontrolled hypertension and subsequent cardiovascular complications. Factors associated with non-adherence to AHMs included a monthly household income below 10,000 Afghanis, multiple daily AHMs, comorbid medical conditions, lack of awareness about hypertension complications, and the presence of depressive symptoms. Given the current socio-economic climate in Afghanistan, with widespread unemployment and the majority living below the poverty line compounded by constraints from international donors, the issue of medication adherence takes on an even graver significance. This situation underscores the need for innovative, locally appropriate, and economically sensitive solutions tailored to the unique challenges of Afghanistan. Thus, policymakers, clinicians, international partners should collaboratively prioritize and implement evidence-based interventions that not only address the specific barriers to AHMs adherence in the country but also provide sustainable, long-term solutions.

## Supporting information

**S1 Dataset. Microsoft excel file with minimal dataset.**
(XLS)

## Acknowledgments

We express our gratitude to the officials in Mirwais Regional Hospital. We offer special thanks to our subjects and data collectors of making this study possible through their generous contribution.

## Author Contributions

**Conceptualization:** Muhammad Haroon Stanikzai, Mohammad Hashim Wafa, Essa Tawfiq, Abdul Wahed Wasiq, Charuai Suwanbamrung.

**Data curation:** Muhammad Haroon Stanikzai, Bilal Ahmad Rahimi.

**Formal analysis:** Bilal Ahmad Rahimi, Ahmad Haroon Baray, Charuai Suwanbamrung.

**Funding acquisition:** Muhammad Haroon Stanikzai, Cua Ngoc Le, Charuai Suwanbamrung.

**Investigation:** Mohammad Hashim Wafa, Essa Tawfiq, Charuai Suwanbamrung.

**Methodology:** Muhammad Haroon Stanikzai, Massoma Jafari, Cua Ngoc Le, Abdul Wahed Wasiq, Ahmad Haroon Baray, Charuai Suwanbamrung.

**Project administration:** Massoma Jafari, Ahmad Haroon Baray, Temesgen Anjulo Ageru, Charuai Suwanbamrung.

**Supervision:** Charuai Suwanbamrung.

**Validation:** Charuai Suwanbamrung.

**Writing – original draft:** Muhammad Haroon Stanikzai, Mohammad Hashim Wafa, Essa Tawfiq, Massoma Jafari, Cua Ngoc Le, Abdul Wahed Wasiq, Bilal Ahmad Rahimi, Temesgen Anjulo Ageru, Charuai Suwanbamrung.

**Writing – review & editing:** Muhammad Haroon Stanikzai, Mohammad Hashim Wafa, Essa Tawfiq, Massoma Jafari, Cua Ngoc Le, Bilal Ahmad Rahimi, Charuai Suwanbamrung.

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
