## [Decision Letter · Decision Letter 0]

5 Oct 2023

PONE-D-23-29407Predictors of non-adherence to antihypertensive medications: A cross-sectional study from a regional hospital in AfghanistanPLOS ONE

Dear Dr. Suwanbamrung,

Thank you for submitting your manuscript to PLOS ONE. After careful consideration, we feel that it has merit but does not fully meet PLOS ONE’s publication criteria as it currently stands. Therefore, we invite you to submit a revised version of the manuscript that addresses the points raised during the review process.

We look forward to receiving your revised manuscript.

Kind regards,

Kahsu Gebrekidan

Academic Editor

PLOS ONE

Additional Editor Comments:

Please address the comments forwarded from both reviewers.

Reviewers' comments:

Reviewer's Responses to Questions

**Comments to the Author**

1. Is the manuscript technically sound, and do the data support the conclusions?

Reviewer #1: Yes

Reviewer #2: Yes

2. Has the statistical analysis been performed appropriately and rigorously? 

Reviewer #1: Yes

Reviewer #2: Yes

3. Have the authors made all data underlying the findings in their manuscript fully available?

Reviewer #1: Yes

Reviewer #2: Yes

4. Is the manuscript presented in an intelligible fashion and written in standard English?

Reviewer #1: Yes

Reviewer #2: Yes

5. Review Comments to the Author

Reviewer #1: Cut off points for assessing depression status is not mentioned in methodology.

Cut-off points for assessing physical activity are not mentioned in methodology.

Class interval for age in Table (1) is not equal.

In Table (4), why 40 is used as cut off point for age.

Where is the p value for table (4)

Reviewer #2: Methodology

The authors should have explained the reliability and validity of the questionnaires used after translation to their local language. I would recommend doing a reliability analysis as a small measure before adding the full analysis of the study.

By translation of the questionnaires used and not stating the reliability this would jeopardize the results of the study.

Results

The authors didn’t consider in multiple medication asking the combined pills strategy. There is new drugs on the market with three active ingredients for hypertension or medication for hypertension. May be this could lead to more adherence to medication rather than multiple medications during the day and night time.

Discussion

The authors didn’t mention drug to drug interaction as a cause of non adherence in the presence of co morbidities.

Overall comment

The study is overall well planned and well written. The authors did a huge effort in combining three questionnaires in this study to get the results of the study.

6. PLOS authors have the option to publish the peer review history of their article (what does this mean?). If published, this will include your full peer review and any attached files.

Reviewer #1: No

Reviewer #2: No

---

## [Author Response · Author response to Decision Letter 0]

15 Oct 2023

Dear Editor,

We would like to thank the editorial board and the reviewers for their thoughtful evaluation of our manuscript entitled “Predictors of non-adherence to antihypertensive medications: A cross-sectional study from a regional hospital in Afghanistan”. Please find our revised manuscript (with tracked changes file), which we believe is substantially strengthened now that we have incorporated reviewers’ recommendations. 

We made amendments according to the journal requirements.

Additional Editor Comments:

Please address the comments forwarded from both reviewers.

Response: Thank you so much. We present our point-by-point response to reviewers’ comments.

Response to Reviewers’ comments

Response to Reviewer 1 comments

1. Cut off points for assessing depression status is not mentioned in methodology.

Response: Thank you so much. We added the details for the cut off points for assessing the depression status in our study. Added details are:

The total scores were grouped into different categories of depression symptoms based on the following ranges: normal= 1–4, mild=5–9, moderate=10–14, severe=15–19, and extremely severe= 20-27 [34,35].

2. Cut-off points for assessing physical activity are not mentioned in methodology.

Response: Thank you. We added the details for the cut off points for assessing the physical activity level in our study.

We considered walking 3 days/week for at least 20 min/day to be low level of physical activity or < 3 metabolic equivalents (METs). Moderate physical activity 3-5days/week for at least 30 min/day represented medium level of physical activity or 3 to 7.9 METs. Vigorous physical activity 3-7 days/week signified high level of physical activity or >8.0 METs [36,37].

3. Class interval for age in Table (1) is not equal. In Table (4), why 40 is used as cut off point for age.

Response: Thank you so much. In table 1, we tried to primarily focus on distribution of age in our sample. In table 4, we focused to categorize age (<40, >40) for a modeling relationship based on relevant literature. This could help us to compare with other studies that have used a cut-off point similar to our one and as well as comparison among young age patients and old age patients.

4. Where is the p value for table (4)?

Response: Thank you so much for this suggestion. We have added p-values for both bivariate and multivariable analysis in Table 4.

Response to Reviewer 2 comments

Methodology

1. The authors should have explained the reliability and validity of the questionnaires used after translation to their local language. I would recommend doing a reliability analysis as a small measure before adding the full analysis of the study.

By translation of the questionnaires used and not stating the reliability this would jeopardize the results of the study.

Response: We thank you for this observation: we agree! We have taken your advice, and for each scale, we added the reliability analysis results to the manuscript. Additionally, we also have provided references for the validity and reliability of translated scales from other studies conducted in Afghanistan.

Results

2. The authors didn’t consider in multiple medication asking the combined pills strategy. There are new drugs on the market with three active ingredients for hypertension or medication for hypertension. May be this could lead to more adherence to medication rather than multiple medications during the day and night time.

Response: We thank you for this helpful comment. We agree that obtaining information about combined pill strategies from hypertensive patients could have offered additional insights into the factors affecting medication adherence. However, it is worth noting that Afghanistan has a low literacy level, and the responses from individuals there may be subject to recall bias. We have included additional information in the discussion section highlighting the significance of combined pill strategies and its preparation called fixed dose combinations as potential contributors to improved adherence.

Added details:

However, regimens with more than one medication per day are inconvenient, and as a consequence, patients find them difficult to follow, and potentially compromising the effectiveness of the treatment. Hence, the combination and preparation of the two AHMs as one tablet, known as a fixed-dose combination (FDC), might improve patient adherence [21,41]. Before prescribing FDCs, healthcare providers should carefully consider the medical history of patients, their current health status, and potential drug interactions.

Discussion

3. The authors didn’t mention drug-to-drug interaction as a cause of non-adherence in the presence of co morbidities.

Response: We thank you for this suggestion. We have added this important suggestion to the discussion section of our manuscript.

Added details:

Another limitation associated with comorbidities is the potential for drug-drug interactions, which can hinder medication adherence. It is crucial for healthcare providers to effectively communicate the risks and benefits of each medication, especially when potential interactions exist [21]. By understanding the significance, patients may be more motivated to adhere to the prescribed regimen and report any unusual side effects promptly [21,40]. Therefore, hypertensive patients with a comorbidity warrant adequate care, supervision and counseling during their treatment. 

Overall comment

The study is overall well planned and well written. The authors did a huge effort in combining three questionnaires in this study to get the results of the study.

Response: We thank you for the constructive feedback. There is a large gap of health research in Afghanistan and we believe this study could be of potential importance for policy and make a baseline for future studies of medication adherence. 

Best Regards,

Corresponding Author

---

## [Decision Letter · Decision Letter 1]

20 Nov 2023

Predictors of non-adherence to antihypertensive medications: A cross-sectional study from a regional hospital in Afghanistan

PONE-D-23-29407R1

Dear Dr. Charuai,

We’re pleased to inform you that your manuscript has been judged scientifically suitable for publication and will be formally accepted for publication once it meets all outstanding technical requirements.

Kind regards,

Kahsu Gebrekidan

Academic Editor

PLOS ONE

Additional Editor Comments (optional):

Reviewers' comments:

Reviewer's Responses to Questions

**Comments to the Author**

1. If the authors have adequately addressed your comments raised in a previous round of review and you feel that this manuscript is now acceptable for publication, you may indicate that here to bypass the “Comments to the Author” section, enter your conflict of interest statement in the “Confidential to Editor” section, and submit your "Accept" recommendation.

Reviewer #1: All comments have been addressed

Reviewer #3: All comments have been addressed

2. Is the manuscript technically sound, and do the data support the conclusions?

Reviewer #1: Yes

Reviewer #3: Yes

3. Has the statistical analysis been performed appropriately and rigorously? 

Reviewer #1: Yes

Reviewer #3: Yes

4. Have the authors made all data underlying the findings in their manuscript fully available?

Reviewer #1: Yes

Reviewer #3: Yes

5. Is the manuscript presented in an intelligible fashion and written in standard English?

Reviewer #1: Yes

Reviewer #3: Yes

6. Review Comments to the Author

Reviewer #1: All the comments have been addressed carefully

1. Cut off points for assessing depression status is not mentioned in methodology.

Response: Thank you so much. We added the details for the cut off points for assessing the depression status in our study. Added details are:

The total scores were grouped into different categories of depression symptoms based on the following ranges: normal= 1–4, mild=5–9, moderate=10–14, severe=15–19, and extremely severe= 20-27 [34,35].

2. Cut-off points for assessing physical activity are not mentioned in methodology.

Response: Thank you. We added the details for the cut off points for assessing the physical activity level in our study.

We considered walking 3 days/week for at least 20 min/day to be low level of physical activity or < 3 metabolic equivalents (METs). Moderate physical activity 3-5days/week for at least 30 min/day represented medium level of physical activity or 3 to 7.9 METs. Vigorous physical activity 3-7 days/week signified high level of physical activity or >8.0 METs [36,37].

3. Class interval for age in Table (1) is not equal. In Table (4), why 40 is used as cut off point for age.

Response: Thank you so much. In table 1, we tried to primarily focus on distribution of age in our sample. In table 4, we focused to categorize age (<40, >40) for a modeling relationship based on relevant literature. This could help us to compare with other studies that have used a cut-off point similar to our one and as well as comparison among young age patients and old age patients.

4. Where is the p value for table (4)?

Response: Thank you so much for this suggestion. We have added p-values for both bivariate and multivariable analysis in Table 4.

Reviewer #3: Thank you for submitting manuscript and your hard work is really appreciated. Manuscript is clear, concise and well written. There is no any additional comments.

7. PLOS authors have the option to publish the peer review history of their article (what does this mean?). If published, this will include your full peer review and any attached files.

Reviewer #1: No

Reviewer #3: No

---

## [Editor Report · Acceptance letter]

15 Dec 2023

PONE-D-23-29407R1 

PLOS ONE

Dear Dr. Suwanbamrung, 

I'm pleased to inform you that your manuscript has been deemed suitable for publication in PLOS ONE. Congratulations! Your manuscript is now being handed over to our production team.

Kind regards, 

on behalf of

Dr. Kahsu Gebrekidan 

Academic Editor

PLOS ONE